# Manipulable preoperative factors affecting surgical outcomes of iStent inject W, particularly the type of antiglaucoma medications

**Sayaka Kimura-Uchida, Ryuichi Shimada, Hiroshi Horiguchi**[ID]*,
**Satoshi Katagiri, Hisato Gunji, Tadashi Nakano**[ID]

Department of Ophthalmology, The Jikei University School of Medicine, Minato-ku, Tokyo, Japan

* hhiro@jikei.ac.jp

## Abstract

To analyze the surgical outcomes of iStent inject W (ISIW) implantation and investigate the influence of preoperative factors. In total, 114 eyes of 114 patients (mean age, 73.22 ± 7.84 y) were enrolled in this retrospective study. The patients underwent ISIW implantation at the Jikei University Hospital. The number of antiglaucoma medications was converted into glaucoma medication scores (GMS). Linear mixed model (LMM) analysis was performed by setting GMS and intraocular pressure (IOP) change as objective variables, and the postoperative days, preoperative mean deviation (MD), preoperative IOP, and various antiglaucoma medications as fixed effects. The preoperative IOP was 15.06 ± 3.51 mmHg and significantly reduced to 12.22 ± 2.23 mmHg at 3 months and 12.99 ± 2.45 mmHg at 12 months. The mean GMS was 2.46 ± 1.33 preoperatively, and decreased to 1.32 ± 1.31 at 3 months, and 1.60 ± 1.41 at 12 months postoperatively. The IOP and GMS values were based on a subset of 72 eyes with 12-months of complete follow-up data. LMM analysis showed significant coefficients for IOP change in relation to postoperative days and preoperative IOP, and for GMS change in relation to postoperative days, β blockers (BB), and Rho kinase inhibitors (ROCK inhibitors). Preoperative factors, particularly medication use, influenced outcomes, indicating that BB or ROCK inhibitors were associated with a greater reduction in the need for postoperative antiglaucoma medications.

## Introduction

Currently, lowering intraocular pressure (IOP) is the only evidence-based treatment for glaucoma [1]. Topical antiglaucoma medications are the main methods to reduce IOP, and surgical treatments have generally been considered in eyes with uncontrolled glaucoma by topical antiglaucoma medications only [2]. Trabeculectomy has the merit of strongly reducing IOP, but it sometimes induces severe complications such as hypotony maculopathy, suprachoroidal hemorrhages, and anterior chamber bleeding or

**Data availability statement:** Data cannot be shared publicly because of ethical restrictions. Data are available from the Institutional Review Board of the Jikei University School of Medicine for researchers who meet the criteria for access to confidential data. The Jikei University Hospital Ethics Committee Secretariat 3-25-8 Nishi-Shimbashi, Minato-ku Tokyo, Japan, 105-8461 TEL:+81-3-3433-1111.

**Funding:** This study was supported by Japan Society for the Promotion of Science (JSPS) KAKENHI (JP21K09729 to H.H.). The funders had no role in study design, data collection and analysis, decision to publish, or preparation of the manuscript. There was no additional external funding received for this study.

**Competing interests:** The present study was supported only by the Japan Society for the Promotion of Science (JSPS KAKENHI, JP21K09729 to H.H.). Dr. Tadashi Nakano has received research funding from Alcon Japan Ltd., Santen Pharmaceutical Co., Ltd., Senju Pharmaceutical Co., Ltd., Otsuka Pharmaceutical Co., Ltd., and Kowa Company, Ltd. within the past five years. Dr. Hiroshi Horiguchi has received research funding from Alcon Japan Ltd. within the past five years. These relationships are unrelated to the present study. There are no patents, products in development or marketed products associated with this research to declare. This does not alter our adherence to PLOS ONE policies on sharing data and materials. The authors declare no other competing interests.

hyphema [3,4]. To improve surgical effects or reduce complications, various glaucoma surgeries, including micro invasive glaucoma surgeries (MIGS) such as Trabectome [5], Hydrus microstent [6–8], Kahook Dual Blade [9], gonioscopy-assisted transluminal trabeculotomy [10], 360 degrees suture trabeculotomy [11] and iStent [12], have been developed. Among these surgeries, the iStent series, released in 2019 and developed up to the third generation (iStent inject W; ISIW), is one of the first MIGS devices used [13] and is known for its minimal invasiveness and low incidence of complications [14]. ISIW is characterized by a larger base that aids in achieving finer positioning during surgery. In addition, ISIW achieves stable IOP reduction through its dual implantation system, which prevents device occlusion with the larger base and provides a direct pathway for aqueous outflow from the anterior chamber to the Schlemm's canal [14–17]. Previous studies have reported good surgical outcomes of ISIW implantation such as IOP-lowering effects of approximately 1.8–2.7 mmHg at 12 months postoperatively and decrease of antiglaucoma medications [14,15,17].

Importantly, nearly all glaucoma patients are treated with antiglaucoma medications before MIGS. The National Institute for Health and Care Excellence and the American Academy of Ophthalmology recommend the prostaglandin analog (PG) as the first choice for glaucoma therapy due to its superiority in terms of IOP-lowering effects compared to other antiglaucoma medications [2]. However, the kinds of antiglaucoma medications were determined by IOP-lowering effects as well as other factors such as the neuroprotection of alpha2-agonist (AA) [18]. In fact, antiglaucoma medications have often been reported to affect surgical outcomes for various pharmacological reasons. Preoperative use of bimatoprost reduces the success rate of trabeculectomy [19]. Preoperative use of ROCK inhibitors leads to better surgical outcomes in eyes that underwent glaucoma surgeries [20–22].

Thus, it is important to clarify which preoperative factors affect surgical outcomes when considering the surgical indications for ISIW implantation. Previous studies have reported that high preoperative IOP led to a significant IOP-lowering effect in ISIW implantation [23] and preoperative central corneal thickness and IOP affected postoperative IOP in second-generation iStents [24,25]. However, there have been no reports on the extent to which preoperative medications could affect the surgical outcomes of ISIW implantation. Notably, preoperative antiglaucoma medications are one of the manipulable factors by ophthalmologists in a clinic, although drug allergies and adherence issues remain. In this study, we analyzed the surgical outcomes of eyes with ISIW implantation and investigated the influence of preoperative factors, including the antiglaucoma medications, using a linear mixed model (LMM).

## Materials and methods

### Study design and ethical considerations

This multicenter retrospective study evaluated the effectiveness of iStent implantation. This study was approved by the Institutional Review Board of the Jikei University School of Medicine and was conducted in accordance with the principles of the Declaration of Helsinki (approval number: 32–307 [10389]). Written informed consent was obtained from all the participants.

Preoperative topical antiglaucoma medications were generally standardized according to the guidelines of the relevant academic societies. However, postoperative topical antiglaucoma medications showed variability as they were left to the discretion of the treating ophthalmologist based on individual patient needs. Given this variability, we focused our analysis primarily on preoperative factors to ensure consistency in this study.

In addition, the progression of glaucoma varies widely among patients, making it clinically challenging to standardize postoperative topical antiglaucoma medications across individuals. To address this, our study adopted a protocol in which preoperative antiglaucoma medications were discontinued on the first postoperative day. Subsequent topical antiglaucoma medications, if needed, were prescribed by the attending ophthalmologist during follow-up visits. This protocol was applied uniformly to all patients to ensure equitable treatment and prioritize patient care.

## Patients selection and data acquisition

This study retrospectively reviewed the medical records of patients who met the inclusion criteria between April 2021 and November 2024. All patients were medicated in four hospitals affiliated with the Jikei University School of Medicine: Jikei University Hospital, Jikei Daisan Hospital, Jikei Kashiwa Hospital, and Jikei Katsushika Hospital. The inclusion criteria were as follows: 1) underwent cataract and glaucoma surgery with ISIW implantation (Glaukos Corporation, Laguna Hills, CA, USA); 2) age over 20 years; 3) follow-up periods of > 3 months after surgery; 4) IOP of 25 mmHg or less under antiglaucoma medications; and 5) no ophthalmic or systemic history that might affect surgical outcomes. When both eyes of the same patient fulfilled these criteria, only the eye with the earlier surgical date was included, in order to avoid duplicate data from the same patient. The inclusion of a placebo group was not feasible in this study because the benefit of MIGS in reducing IOP has already been well established, and intentionally creating a placebo group for patients with glaucoma would not have been ethically acceptable. Moreover, introducing a placebo group would have required an appropriately larger sample size, especially considering the number of fixed effects included in the LMM analysis; however, the number of eligible patients who did not wish to undergo MIGS was too small to allow for meaningful comparison or randomization.

The following data were recorded for all patients; age, sex, eye laterality, glaucoma type, IOP, pre- and postoperative antiglaucoma medications, preoperative visual field, and surgical records. IOP was measured using Goldmann applanation tonometry, and the preoperative IOP was determined as the average of two measurements at different times. The preoperative visual fields were assessed using a Humphrey Visual Field Analyzer. The usage history of pre- and postoperative antiglaucoma medications was converted to a glaucoma medication score (GMS) according to previous studies [14,15,19,25,26]. GMS was the total score calculated by counting each single-agent antiglaucoma medication as one point and each combination antiglaucoma medication as two points, based on the types of antiglaucoma medications used. Postoperative IOP and GMS data were obtained at 1 and 7 days, and 1, 3, 6, 9, and 12 months after surgery. As our hospitals are university referral centers, many patients had already started topical antiglaucoma medications at the time of referral, and untreated baseline IOP data were not available. Therefore, the exact proportion of patients with normal tension glaucoma (NTG) could not be determined. As a result, the changes in IOP and GMS were set to be the main outcome to evaluate the effects of ISIW implantation.

## ISIW implantation with cataract surgery

All patients underwent third-generation ISIW implantation following phacoemulsification cataract surgery under topical and intracameral anesthesia (4% and 0.5% lidocaine, Xylocaine; Sandoz Pharma KK, Tokyo, Japan) after confirming the absence of metal allergy and meeting the criteria for ISIW usage from the Japanese Ophthalmological Society (Japanese Guidelines for Use of iStent or iStent Inject Trabecular Micro Bypass in Combination with Cataract Surgery, 2nd Edition). After dilating the pupil, a main 2.4-mm clear corneal incision was made along the steep axis of the cornea, and one or two clear corneal incisions were made across the main clear corneal incision. The anterior chamber was filled with viscoelastic material (Healon, AMO Japan, Tokyo, Japan), and continuous curvilinear capsulorhexis was performed. After

phacoemulsification, a one-piece soft acrylic intraocular lens was inserted through the main corneal incision. The anterior chamber was filled with viscoelastic material (Shellgan, Santen Pharmaceutical, Osaka, Japan), and two ISIW stents were implanted in the nasal region of the Schlemm's canal at the 8–10 o'clock positions in the left eye and the 2–4 o'clock positions in the right eye using a gonioprism lens (Ocular Instruments, Swan-Jacob Lens, Bellevue, WA, USA). The ISIW was implanted through either the main clear corneal incision (for against-the-rule astigmatism) or the temporal clear corneal incision (for with-the-rule astigmatism). After the ISIW implantation, the viscoelastic material was irrigated from the anterior chamber using a balanced salt solution. At the end of the surgery, steroids (6.6 mg/vial dexamethasone; Dexart, Fuji Pharma, Tokyo, Japan) and an antibiotic ointment (0.3% topical ofloxacin; Tarivid, Santen Pharmaceutical, Osaka, Japan) were administered. Postoperative topical therapy included antibiotics (1.5% levofloxacin; Santen Pharmaceutical, Osaka, Japan), steroids (0.1% dexamethasone; Orgadrone, Sandoz Pharma KK, Japan), and NSAIDs (0.1% diclofenac; Diclod, Wakamoto Pharmaceutical, Tokyo, Japan). Further use of topical antiglaucoma medications was at the discretion of the attending ophthalmologist [15,27].

## Data analysis

We compared the pre- and post-operative IOP and GMS using Tukey's HSD test for paired samples. To address potential bias from the inclusion of multiple time points for the same patient, we employed LMM analysis using the R packages lme4, lmerTest, MuMIn, and emmeans to assess changes in GMS and IOP. Additionally, to minimize intraindividual bias, only one eye per patient was included in the final analysis when both eyes met the inclusion criteria. The LMM can account for the fact that eyes may have different IOP and GMS values over time within the same individual [17,28,29]. For both analyses, the GMS or IOP change at 3, 6, 9, and 12 months after surgery was set as the objective variable. Changes in IOP and GMS were defined as the preoperative value minus the postoperative value at each time point. Therefore, a positive coefficient indicated that the variable was associated with a greater reduction in postoperative IOP or GMS. Postoperative days, preoperative visual field mean deviation (MD), preoperative IOP, and various antiglaucoma medications were set as fixed effects. Therefore, individual patients (ID) were set as random intercepts to account for intraindividual correlations arising from the inclusion of data from multiple time points for the same patient.

To address concerns that the LMM results might be influenced by the preferences or decisions of attending doctors, we conducted additional analyses by including both individual patients and each type of antiglaucoma medication as random intercepts. The analysis revealed minimal differences in the random intercepts for individual antiglaucoma medications, indicating that specific medications were not preferentially prescribed according to attending physicians' preferences. To further investigate the effects and interactions of each medication, we included preoperative antiglaucoma medication types as fixed effects in the LMM. Previous studies have reported that IOP mostly decreases within 1–3 months after ISIW implantation, while GMS reductions are mostly prominent during the first observation after surgery [15–17,26]. Therefore, in our LMM analysis, the primary outcome was the use of postoperative data from 3 to 12 months. The threshold for statistical significance was set at $p < 0.05$. All statistical analyses were performed using Python (ver. 3.10.9) and R (Ver. 4.3.1).

## Results

### Patient demographics

We retrospectively reviewed the medical records of 114 eyes of 114 patients who underwent ISIW implantation. Pre- and postoperative IOP and GMS data up to 3 months after surgery were obtained for all participants (100%). At 6, 9, and 12 months after surgery, IOP data were available for 97 eyes of 97 patients (85%), 84 eyes of 84 patients (74%), and 75 eyes of 75 patients (66%), respectively. GMS data were available for 98 eyes of 98 patients (86%), 85 eyes of 85 patients (75%), and 77 eyes of 77 patients (68%), respectively.

In 114 eyes of 114 patients, the examined (operated) age ranged from 50 to 89 years (mean±SD; 73.22±7.84 years). Open-angle glaucoma, which included primary open-angle and normal-tension glaucoma, affected 104 eyes of 104 patients (91.2%), and pseudoexfoliation glaucoma affected 10 eyes of 10 patients (8.8%). Because many patients were referred from other clinics and had already started topical antiglaucoma medications, untreated baseline IOP data were not available. Consequently, the exact proportion of patients with normal-tension glaucoma could not be determined. Furthermore, 49 (43.0%) patients were women and 65 (57.0%) were men. There were 66 (57.9%) right eyes and 48 (42.1%) left eyes. Demographic data are summarized in Table 1.

## Preoperative antiglaucoma medications

For preoperative glaucoma therapy, 27 (23.7%) eyes were medicated by PG only; 15 (13.2%) eyes by the PG, BB, carbonic anhydrase inhibitor (CAI), and AA; 13 (11.4%) eyes by PG and BB; and 10 eyes (8.8%) medicated PG, BB, CAI, AA, and ROCK inhibitor (Fig 1A). The numbers of other antiglaucoma medications administered are summarized in Fig 1A. In other words, PG was used in 103 (90.4%) eyes, BB in 64 (56.1%) eyes, other medications in 69 (60.5%) eyes (Fig 1B), CAI in 51 (44.7%) eyes, AA in 45 (39.5%) eyes, and ROCK inhibitor in 19 (16.7%) eyes.

## Pre- and post-operative changes of IOP and GMS

The preoperative IOP and GMS were 15.06±3.51 mmHg and 2.46±1.33, respectively. The preoperative MD (mean±SD) was −6.79±3.94 dB. The postoperative IOP was 14.40±5.38 mmHg the next day, 16.53±4.74 mmHg 7 days later, and 14.12±3.35 mmHg 1 month later. Compared to preoperative values, postoperative IOP decreased significantly to 12.22±2.23 mmHg 3 months later, 12.92±2.73 mmHg 6 months later, 12.81±2.64 mmHg 9 months later, and 12.99±2.45 mmHg 12 months later (p<0.05) (Fig 2A). Postoperative GMS decreased significantly compared to preoperative values to 0.00±0.00 the next day, 0.74±1.29 7 days later, 1.15±1.11 1 month later, 1.32±1.31 3 months later, 1.39±1.34 6 months later, 1.40±1.38 9 months later, and 1.60±1.41 12 months later (P<0.05) (Fig 2B). The analyses were performed using data from 72 eyes of 72 patients over a 12-month follow-up period.

**Table 1. Demographic and clinical data of patients undergoing ISIW implantation.**

| Variables | Values |
| --- | --- |
| No. eyes | 114 |
| No. patients | 114 |
| Age, mean±SD, years | 73.22±7.84 |
| Glaucoma type | |
| POAG, n (%) | 104 (91.23) |
| PEG, n (%) | 10 (8.77) |
| Gender | |
| Male, n (%) | 65(57.02) |
| Female, n (%) | 49 (42.98) |
| Right and left | |
| Right, n (%) | 66 (57.89) |
| Left, n (%) | 48 (42.11) |
| Baseline | |
| IOP, mean±SD, mmHg | 15.43±3.61 |
| GMS, mean±SD | 2.47±1.33 |
| MD, dB | −7.52±4.81 |

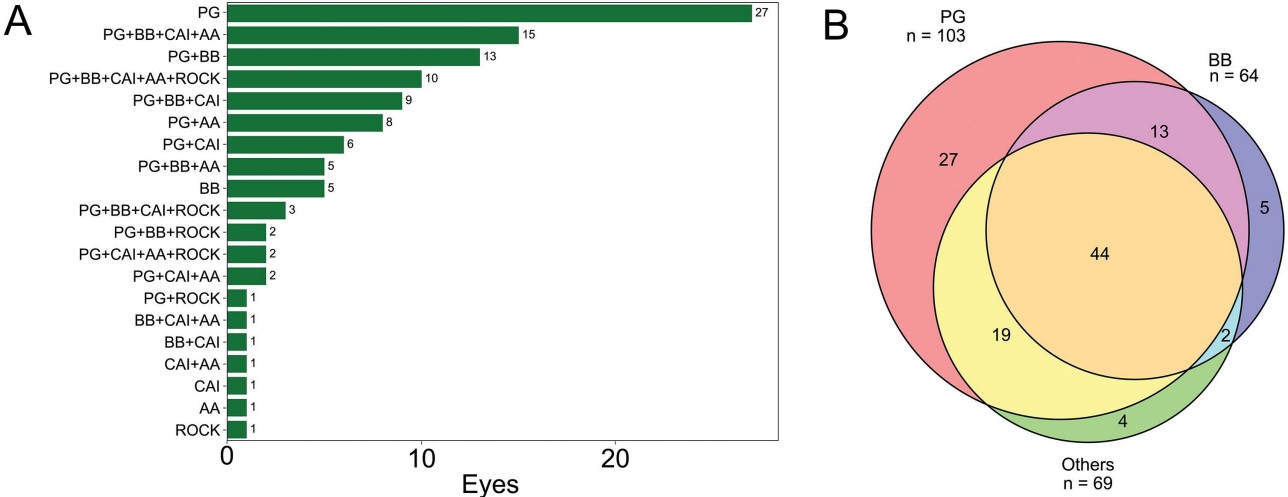

**Fig 1. Patterns and frequency of preoperative antiglaucoma medications.** The data of 114 eyes are shown by a horizontal bar graph (A) and a Venn diagram (B). PG is most frequently used in 103 eyes (90.4%) including 27 eyes as single patterns and 76 eyes as combination patterns. For other types of antiglaucoma medications, BB is used in 64 eyes (56.1%), CAI in 51 eyes (44.7%), AA in 45 eyes (39.5%), and ROCK inhibitor in 19 eyes (16.7%).

## LMM analysis of the relationship between preoperative factors and surgical outcomes

IOP changes were measured in 114 eyes of 114 patients, 97 eyes of 97 patients, 84 eyes of 84 patients, and 75 eyes of 75 patients at 3, 6, 9, and 12 months after the surgery, respectively. GMS changes were measured in 114 eyes of 114 patients, 98 eyes of 98 patients, 85 eyes of 85 patients, and 77 eyes of 77 patients at 3, 6, 9, and 12 months after surgery, respectively. These data were used for LMM analysis.

The LMM for IOP change explained approximately 43% of the data by fixed effects and 76% by random effects. The value of coefficient was $-0.00284 \pm 0.000557$ for postoperative days ($p < 0.01$), $-0.0150 \pm 0.0365$ in preoperative MD values ($p = 0.682$), $0.533 \pm 0.0496$ in preoperative IOP values ($p < 0.01$), $-0.262 \pm 0.639$ in PG values ($p = 0.683$), $0.136 \pm 0.369$ in BB values ($p = 0.713$), $-0.0401 \pm 0.398$ in CAI values ($p = 0.920$), $-0.162 \pm 0.380$ in AA values ($p = 0.670$), and $0.136 \pm 0.481$ in ROCK inhibitor values ($p = 0.778$). Postoperative days and preoperative high IOP were significant factors affecting postoperative IOP change. The data are summarized in Table 2.

The LMM for the GMS change explained approximately 26% of the data using fixed effects and 89% using random effects. The value of the coefficient was $-0.000817 \pm 0.000186$ in postoperative days values ($p < 0.01$), $-0.0293 \pm 0.0235$ in preoperative MD values ($p = 0.215$), $-0.0452 \pm 0.0322$ in preoperative IOP values ($p = 0.163$), $0.310 \pm 0.421$ in PG values ($p = 0.464$), $0.719 \pm 0.240$ in BB values ($p < 0.01$), $0.434 \pm 0.259$ in CAI values ($p = 0.0976$), $0.292 \pm 0.247$ in AA values ($p = 0.240$), and $0.734 \pm 0.313$ in ROCK inhibitor values ($p < 0.05$). Postoperative days significantly decreased postoperative GMS changes while preoperative BB and ROCK inhibitor use significantly increased postoperative GMS changes. The data are summarized in Table 3.

To further contextualize the clinical meaningfulness of postoperative medication reduction, we performed an exploratory analysis. "Medication-free" was defined as GMS = 0 at 12 months, and the absolute risk difference (ARD) in medication-free rates was calculated as P(med-free | medication use) − P(med-free | no medication use). NNT (Number Needed to Treat) is only shown when ARD > 0. As shown in S1 Table, the medication-free rate at 12 months was lower in patients preoperatively treated with beta-blockers (21.4% vs. 33.3%, ARD = −11.9%, 95% CI: −32.3% to 8.3%), and higher in those treated with ROCK inhibitors (36.4% vs. 24.6%, ARD = +11.8%, 95% CI: −12.6% to 41.5%).

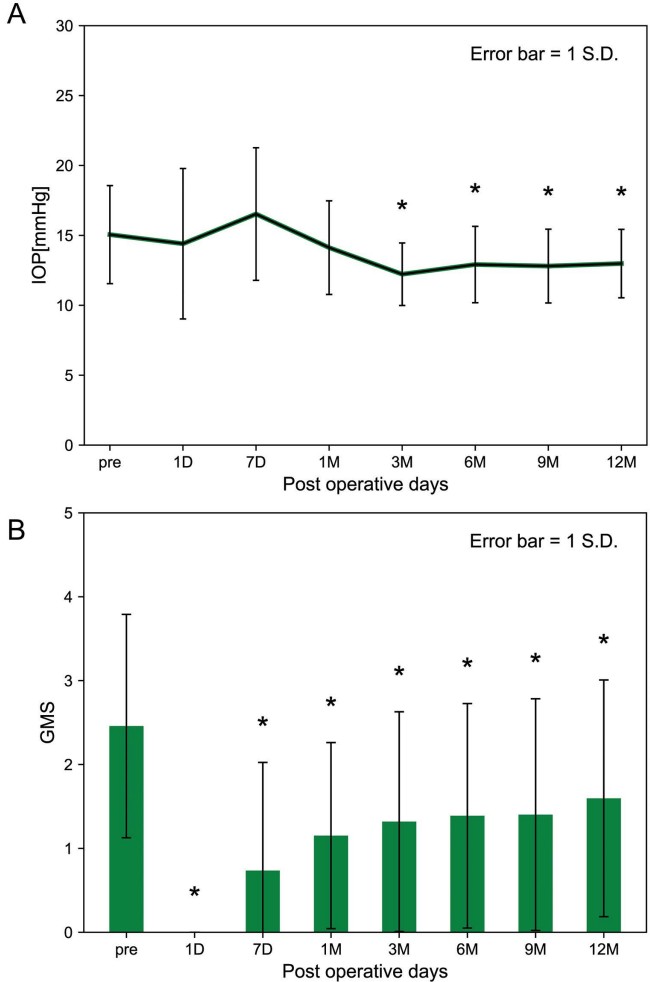

**Fig 2. Surgical changes through ISIW implantation.** The data of 72 eyes are shown. Vertical axis represents IOP (A) and GMS (B), respectively. Horizontal axis represents time from preoperative timing to 12 months after surgery. Pre- and postoperative data are compared by Tukey's HSD test and asterisks mean statistical significance (p<0.05). IOP decreases significantly from 3 to 12 months after surgery whereas GMS decreases from one day to 12 months after surgery. D, days; GMS, glaucoma medication score; IOP, intraocular pressure; M, months.

## Discussion

According to the guidelines of the World Glaucoma Association, the American Academy of Ophthalmology, and the Japan Glaucoma Society, PG is recommended as the first-line therapy for glaucoma because of its superior IOP-lowering effects [30–32]. When monotherapy is insufficient, adjunctive treatment with BB or EP2 receptor agonists is typically recommended. This can be followed by additional agents if required, such as CAI, AA, or ROCK inhibitors, depending on the clinical context [32]. In our study, nearly all patients were prescribed PG or BB as the initial treatment. When sufficient IOP reduction was not achieved or when tolerance or adverse effects occurred with these medications, additional medications from other classes were added or substituted in a stepwise manner. This guideline-based approach explains the variety of antiglaucoma medication combinations shown in Fig 1A and 1B.

In this study, we confirmed a reduction in IOP and GMS in eyes with ISIW implantation until 12 months after surgery, which is consistent with previous reports. Furthermore, we evaluated how much the preoperative factors including

Table 2. Results of the LMM analysis for IOP change.

| Variables | Coefficients ± SE | P value | 95% CI used | not used |
|---|---|---|---|---|
| | **Values** | | | |
| | | | **95% CI** | |
| Postoperative days, days | −0.00284 ± 0.000557 | < 0.01 | | |
| Preop MD, dB | −0.0150 ± 0.0365 | 0.682 | | |
| PreIOP, mmHg | 0.533 ± 0.0495 | < 0.01 | | |
| Glaucoma agents | | | | |
| PG | −0.262 ± 0.639 | 0.683 | 1.91-2.85 | 1.37-3.90 |
| BB | 0.136 ± 0.369 | 0.713 | 1.82-3.33 | 1.59-3.28 |
| CAI | −0.0401 ± 0.398 | 0.920 | 1.67-3.31 | 1.72-3.34 |
| AA | −0.162 ± 0.380 | 0.670 | 1.59-3.26 | 1.81-3.36 |
| ROCK inhibitor | 0.136 ± 0.481 | 0.778 | 1.56-3.59 | 1.77-3.11 |

Table 3. Results of the LMM analysis for GMS change.

| Variables | Coefficients ± SE | P value | 95% CI used | not used |
|---|---|---|---|---|
| | **Values** | | | |
| | | | **95% CI** | |
| Postoperative days, days | −0.000817 ± 0.000186 | < 0.01 | | |
| Preop MD, dB | −0.0293 ± 0.0235 | 0.215 | | |
| PreIOP, mmHg | −0.0452 ± 0.0322 | 0.163 | | |
| Glaucoma agents | | | | |
| PG | 0.310 ± 0.421 | 0.464 | 1.13-1.74 | 0.29-1.95 |
| BB | 0.719 ± 0.240 | < 0.01 | 1.14-2.13 | 0.37-1.47 |
| CAI | 0.434 ± 0.259 | 0.0976 | 0.96-2.03 | 0.53-1.59 |
| AA | 0.292 ± 0.247 | 0.240 | 0.88-1.97 | 0.62-1.64 |
| ROCK inhibitor | 0.734 ± 0.313 | < 0.05 | 0.98-2.30 | 0.47-1.35 |

antiglaucoma medications, which was one of the manipulable factors by ophthalmologists, affected surgical outcomes using LMM analysis. The difference between the variance explained by the fixed effects (Marginal R²) and the combined fixed and random effects (Conditional R²) in our LMM analysis suggests inter-individual variability in surgical outcomes. While it is essential to provide a treatment that accounts for such individual differences, our results also suggest that the preoperative use of two types of antiglaucoma medications (BB or ROCK inhibitor) could reduce the number of antiglaucoma medications required after ISIW implantation.

LMM analysis was performed using the change in GMS and IOP as objective variables, calculated as the difference between the preoperative and postoperative value (preoperative minus postoperative) at 3–12 months after surgery. This analysis showed a negative coefficient value for postoperative days, although the value was apparently small compared with other variables. The results indicated that the surgical effects of IOP and GMS lowering reduced gradually until one year after surgery. Previous studies, which analyzed 32–90 eyes with ISIW implantation, have also reported that IOP and GMS slowly increased after their peak lowering effect, with observation periods of up to 1 day to 12 months after surgery [15–17,26]. The LMM analysis also showed a positive coefficient for preoperative IOP. Thus, our surgical outcomes of ISIW implantation resulted in better improvement of postoperative IOP in eyes with higher preoperative IOP, which was consistent with previous reports on iStent [23–25]. Furthermore, the Early Manifest Glaucoma Trial demonstrated that

each 1 mmHg reduction in IOP from baseline was associated with an approximately 10% decrease in the risk of glaucoma progression [33]. This indicates that even relatively small IOP reductions may be clinically relevant. Nevertheless, our study design does not allow for a strong discussion of IOP reduction because it allows for changes in the type of antiglaucoma medications based on the visual field and IOP.

Regarding preoperative antiglaucoma medications as one of the manipulable factors by ophthalmologists, the LMM analysis for postoperative GMS showed a statistically significant positive coefficient for BB and ROCK inhibitor. Among preoperative antiglaucoma medications, several studies have reported that the preoperative use of ROCK inhibitors has resulted in better surgical outcomes in eyes undergoing glaucoma surgeries [20–22]. Notably, one study reported higher effectiveness of trabeculotomy in ROCK inhibitor–effective eyes compared to other eyes in the study [20], and another study showed greater success of selective laser trabeculoplasty in ROCK inhibitor–effective eyes compared to other eyes in the study [21]. Furthermore, a separate study reported a lower necessity of additional glaucoma surgeries after microhook ab interno trabeculotomy in the ROCK inhibitor-used group [22]. ROCK inhibitors have pharmacological effects such as relieving outflow resistance in the trabecular meshwork, which enhances aqueous humor drainage through the pathway [34,35], and inhibiting fibroblast proliferation, which helps reduce scarring post-surgery [36]. These effects might have contributed to improved surgical outcomes after ISIW implantation, which targets the trabecular outflow pathway, similar to ROCK inhibitors. We also reevaluated our dataset and found that the mean preoperative IOP was slightly higher in all antiglaucoma medication groups than in the overall cohort. Specifically, the mean preoperative IOP was 15.46 mmHg in the PG group (n = 103), 15.94 mmHg in the BB group (n = 64), 16.08 mmHg in the CAI group (n = 51), 16.22 mmHg in the AA group (n = 45), and 16.37 mmHg in the ROCK inhibitor group (n = 19), while the overall mean IOP was 15.43 mmHg. Although all groups showed slightly higher preoperative IOPs than the overall average, only BB and ROCK inhibitor use was statistically significant in the LMM analysis for GMS changes. This discrepancy suggests that preoperative IOP alone does not fully explain the surgical outcomes. No clinical reports have directly indicated a relationship between the use of BB and the effects of surgical treatments associated with conventional outflow pathways, including laser therapy. Interestingly, when used in combination with PG, BB may have some indirect effects on the trabecular meshwork, leading to lower expression of pro-fibrotic genes in the trabecular meshwork compared to either drug alone [37]. In our study following the glaucoma treatment protocol in Japan, the BB was almost always prescribed together with the PG (Fig 1), which might have produced this effect. Alternatively, the fact that the BB does not produce structural changes in the Schlemm's canal, unlike other glaucoma agents, might have affected surgical outcomes of ISIW implantations.

Our study has several limitations, particularly regarding the strategy of administering preoperative antiglaucoma medications. As shown in Fig 1B, PG was used in 90% of eyes, whereas the ROCK inhibitor was used in 17% of eyes. Clinically, we used PG as the first choice and other antiglaucoma medications as the second or subsequent choices. As a result, BB, AA, CAI, and ROCK inhibitors tended to be used in eyes with more advanced glaucoma, which may have led to a higher preoperative GMS and IOP, as well as greater variation after ISIW implantation.

Additionally, Okuda et al. reported that the duration of preoperative topical antiglaucoma medication use affects surgical outcomes [38]. Since our study was retrospective, and most patients were referred from local clinics to our hospital due to it being a regional core center, we could not obtain accurate information on the onset and duration of preoperative antiglaucoma medication use or determine the extent to which IOP had already been lowered through the use of these medications prior to surgery. We also attempted an additional analysis, including the preoperative GMS (PreGMS) as a fixed effect, along with individual medication variables. However, because the PreGMS represents the sum of all preoperative medications, it was strongly correlated with each individual drug variable, leading to multicollinearity and unstable model estimates. Therefore, PreGMS was not included in the final model.

Furthermore, in this study, patients with 57 eyes used a combination of PG and BB preoperatively, and among them, 32 did not have BB reintroduced at 12 months postoperatively. In the LMM analysis for GMS, BB also showed statistical significance; however, it cannot be ruled out that this reflects the fact that, in these cases, the postoperative IOP reduction

achieved by ISIW eliminated the need to reintroduce BB. Although our study identified a statistically significant association between preoperative BB use and GMS improvement after ISIW implantation, there is currently a lack of direct clinical evidence to evaluate the relationship between BB use and surgical outcomes in MIGS. Accordingly, the observed significance of BB use in our LMM analysis should be interpreted with caution as it may reflect an associative rather than a causal effect. Our exploratory ARD analysis of the GMS indicated that beta-blocker use was associated with a lower likelihood of achieving medication-free status, whereas ROCK inhibitor use was associated with a higher likelihood, although the wide confidence intervals highlight the uncertainty of these findings. These results suggest that, although the absolute differences may appear numerically small, they may still have potential clinical implications for long-term glaucoma management. Thus, further prospective studies with detailed longitudinal tracking of the duration of preoperative medication use are necessary to better inform clinical decision making.

In conclusion, we demonstrated better surgical outcomes with ISIW implantation in terms of the IOP- and GMS-lowering effects in eyes with progressive glaucoma. Furthermore, our analysis revealed that the two types of preoperative antiglaucoma medications, BB or ROCK inhibitors, might amplify the effects of ISIW implantation.

## Supporting information

**S1 Table. Medication-free rates and ARD at 12 months according to preoperative medication use.** "Medication-free" was defined as GMS = 0 at 12 months. ARD was calculated as P(med-free | medication use) – P(med-free | no medication use). NNT (Number Needed to Treat) is only shown when ARD > 0. CI = confidence interval.
(DOCX)

## Acknowledgments

We thank Sho Takahashi for technical support and statistical evaluation, Chiharu Koike, Chihiro Kaizuka, Daisuke Shinohara, Hiroyuki Sasano, Ido Nishijima, Kaori Yamashita, Koki Honzawa, Mari Yamazaki, Masaki Nakamura, Masanobu Iida, Rio Tanaka, Ryo Terauchi, Ryosuke Ito, Shumpei Ogawa, Teruaki Tokuhisa, Tomoyuki Watanabe, Yoshiko Yamawaki, Yoshinori Ito and Yuka Saito for the iStent inject W implantation and following the patients, Ido Nishijima, Shumpei Ogawa and Takahiko Noro for useful comments for preliminary analysis.

## Author contributions

**Conceptualization:** Hiroshi Horiguchi, Hisato Gunji.

**Data curation:** Sayaka Kimura-Uchida, Ryuichi Shimada, Hiroshi Horiguchi.

**Formal analysis:** Sayaka Kimura-Uchida, Ryuichi Shimada, Hiroshi Horiguchi.

**Funding acquisition:** Hiroshi Horiguchi, Hisato Gunji, Tadashi Nakano.

**Investigation:** Sayaka Kimura-Uchida, Hiroshi Horiguchi, Satoshi Katagiri.

**Methodology:** Hiroshi Horiguchi.

**Project administration:** Hiroshi Horiguchi, Hisato Gunji.

**Resources:** Hisato Gunji, Tadashi Nakano.

**Software:** Ryuichi Shimada, Hiroshi Horiguchi.

**Supervision:** Hisato Gunji, Tadashi Nakano.

**Validation:** Ryuichi Shimada.

**Writing – original draft:** Sayaka Kimura-Uchida, Satoshi Katagiri.

**Writing – review & editing:** Hiroshi Horiguchi, Hisato Gunji, Tadashi Nakano.

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
