## [Decision Letter · Decision Letter 0]

24 Jun 2025

Dear Dr. Horiguchi,

Thank you for submitting your manuscript to PLOS ONE. After careful consideration, we feel that it has merit but does not fully meet PLOS ONE’s publication criteria as it currently stands. Therefore, we invite you to submit a revised version of the manuscript that addresses the points raised during the review process.

We look forward to receiving your revised manuscript.

Kind regards,

Yalong Dang

Academic Editor

PLOS ONE

Journal Requirements:

“This study was supported by Japan Society for the Promotion of Science (JSPS) KAKENHI (JP21K09729 to H.H.)”

“This study was supported by Japan Society for the Promotion of Science (JSPS) KAKENHI (JP21K09729 to H.H.)”

“Tadashi Nakano reports grants from All Nippon Airway, Carl Zeiss Meditec AG, HOYA Corporation, IOL MEDICAL, Japan Airlines, Kowa Pharmaceutical, KURIBARA MEDICAL INSTRUMENTS, Kyowa Medical, Otsuka Pharmaceutical, Sanofi, S.A., Senju Pharmaceutical, and Santen Pharmaceutical.

Hiroshi Horiguchi reports grants from Alcon Japan Ltd. Sayaka Kimura-Uchida, Ryuichi Shimada, Satoshi Katagiri and Hisato Gunji have no financial disclosures.”

5. In the online submission form, you indicated that [The data that support the findings of this study are not openly available due to reasons of sensitivity and are available from corresponding author upon reasonable request.].

Reviewers' comments:

Reviewer's Responses to Questions

**Comments to the Author**

1. Is the manuscript technically sound, and do the data support the conclusions?

Reviewer #1: No

Reviewer #2: Partly

2. Has the statistical analysis been performed appropriately and rigorously?

Reviewer #1: Yes

Reviewer #2: I Don't Know

3. Have the authors made all data underlying the findings in their manuscript fully available?

Reviewer #1: Yes

Reviewer #2: No

4. Is the manuscript presented in an intelligible fashion and written in standard English?

Reviewer #1: Yes

Reviewer #2: Yes

Reviewer #1: 1. Study design and ethical considerations;

It is stated that “Preoperative topical antiglaucoma medications were generally standardized according to the guidelines of relevant academic societies,” but given this, isn't the number of combinations of eye drops shown in Figure 1 excessive? An overview of the guidelines should be provided.

2. LMM analysis of the relationship between preoperative factors and surgical outcomes;

Fixed effects and random effects are shown, but their values differ greatly. How should this result be interpreted? In other words, what does the large difference in percentages mean, and how should the results be interpreted?

3. Discussion;

It is stated that “Regarding preoperative antiglaucoma medications as one of the manipulatable factors by ophthalmologists, the LMM analysis for postoperative GMS showed a statistically significant positive coefficient for BB and ROCK inhibitor,” however, the intraocular pressure-lowering effects of each eye drop, the duration of use, and the order of administration are unclear. Based on these results, it is unclear how these eye drops should be used, and whether these results are clinically meaningful.

4. Discussion;

Although it is stated that “when used in combination with PG, BB may have some indirect effects on the trabecular meshwork, leading to lower expression of pro-fibrotic genes in the trabecular meshwork compared to either drug alone.[33],” since iStent forms a bypass with the Schlemm's canal, changes in TM may not be very relevant. In other words, the discussion of PG + BB does not explain why BB was useful. There is insufficient discussion regarding the usefulness of BB.

Reviewer #2: Summary:

This retrospective clinical analysis is aimed to investigate the effect of preoperative anti-glaucoma medications on post-surgical outcomes following iStent inject W implantation. Authors provide relevant context for this investigation in the introduction, describe patient selection criteria, data acquisition, data analysis, and outcomes. GMS change is suggested in relation to preoperative prescription of β blockers and Rho kinase inhibitors. Recommended points to address in the manuscript are included below.

Major points:

1) In the introduction, authors include that “Topical antiglaucoma medications are the main methods to reduce IOP, and surgical treatments have generally been considered in eyes with uncontrolled glaucoma by topical antiglaucoma medications only.[2]”

Authors additionally include that preoperative IOP was an average of approximately 15mmHg.

Is this not considered well-controlled IOP? Why proceed with surgery?

Did a substantial number of these patients have NTG? If there was a substantial number of patients with NTG, was this glaucoma type related to any differences in outcomes?

2) The meaning of this statement is unclear:

“The sample size was insufficient, as it included only patients who did not voluntarily request MIGS.” Insufficient to determine what and insufficient in what way? Authors make conclusions at the end of the study. Are these conclusions supported by the sample size?

3) Authors include in the Discussion,

“ROCK inhibitors have pharmacological effects such as relieving outflow resistance in the trabecular meshwork, which enhances aqueous humor drainage through the pathway,[30, 31] and inhibiting fibroblast proliferation, which helps reduce scarring post-surgery.[32] These effects might have contributed to improved surgical outcomes after ISIW implantation. No clinical reports have directly indicated a relationship between the use of BB and the effects of surgical treatments associated with conventional outflow pathways, including laser therapy. Interestingly, when used in combination with PG, BB may have some indirect effects on the trabecular meshwork, leading to lower expression of pro-fibrotic genes in the trabecular meshwork compared to either drug alone.[33]”

This text seems to suggest that ROCK inhibitors and beta blockers are likely to be helpful as preoperative medications to decrease IOP and/or GMS. However, it seems that they had no significant effect on postoperative IOP and caused significant increases (not improvement) in postoperative GMS.

Additionally, in the abstract, authors include,

“Preoperative factors, particularly medication use, influenced outcomes, indicating that β blockers or Rho kinase inhibitors may reduce the need for postoperative antiglaucoma medications.”

Based on the data shown, it seems that β blockers or Rho kinase inhibitors may increase (not reduce) the need for postoperative antiglaucoma medications. Is this correct? If so, please adjust the manuscript to clarify and justify the increase rather than decrease of GMS as a result of preoperative ROCK inhibitors.

4) Because PLOS one has a broad readership, can authors discuss how clinically meaningful the IOP/GMS changes these preoperative factors may allow is?

For example, how much might the average IOP change of ~0.5mmHg associated with preoperative IOP tend to affect outcomes clinically? How much might a GMS change of ~0.7 (associated with beta blockers) affect outcomes clinically?

This can help contextualize the magnitude of findings.

5) Baseline IOP in Table 1 is different from preoperative IOP in the abstract and results (page 10, first sentence.)

Why is this? Is “baseline” and “preoperative” the same category or different categories?

Minor points:

1) Could authors provide comment on whether or not preoperative medications are likely to affect surgical outcomes after MIGS other than ISIW? Is there anything in particular about the mechanism of ISIW that is likely to be influenced differently by these preoperative medications compared to other iStents or other MIGS?

2) Tables 2 and 3 have the same titles. Is Table 3 intended to signify changes in GMS?

3) The following sentence is a fragment. Please correct this incomplete sentence or combine with the words in the next sentence.

"To improve surgical effects or reduce complications, variable glaucoma surgeries, including micro invasive glaucoma surgeries (MIGS) such as Trabectome,[5] Hydrus microstent,[6-8] Kahook Dual Blade.[9]

"suture, trabeculotomy[11] and iStent,[12] have been developed."

4) The meaning of this statement is unclear:

“When both eyes of the same patient fulfilled these criteria, we used data from an earlier surgical date.”

Data from which eye was used when both eyes of the same patient fulfilled the criteria?

5) Please specify the Python package(s) used for LMM analysis.

6) Please use consistent terms to refer to beta blockers throughout the text (BB vs. β blocker) and ROCK inhibitors / Rho kinase inhibitors.

**Do you want your identity to be public for this peer review?** For information about this choice, including consent withdrawal, please see our Privacy Policy

Reviewer #1: No

Reviewer #2: No

---

## [Author Response · Author response to Decision Letter 1]

14 Sep 2025

Please read an attached word document, "Response to Reviewers".

---

## [Decision Letter · Decision Letter 1]

22 Sep 2025

Manipulable preoperative factors affecting surgical outcomes of iStent inject W, particularly the type of antiglaucoma medications

PONE-D-25-11971R1

Dear Dr. Horiguchi,

We’re pleased to inform you that your manuscript has been judged scientifically suitable for publication and will be formally accepted for publication once it meets all outstanding technical requirements.

Kind regards,

Yalong Dang

Academic Editor

PLOS ONE

Additional Editor Comments (optional):

Reviewer #1:

Reviewers' comments:

Reviewer's Responses to Questions

**Comments to the Author**

Reviewer #1: All comments have been addressed

2. Is the manuscript technically sound, and do the data support the conclusions?

Reviewer #1: Yes

3. Has the statistical analysis been performed appropriately and rigorously?

Reviewer #1: Yes

4. Have the authors made all data underlying the findings in their manuscript fully available?

Reviewer #1: Yes

5. Is the manuscript presented in an intelligible fashion and written in standard English?

Reviewer #1: Yes

Reviewer #1: (No Response)

**Do you want your identity to be public for this peer review?** For information about this choice, including consent withdrawal, please see our Privacy Policy

Reviewer #1: No

---

## [Editor Report · Acceptance letter]

PONE-D-25-11971R1

PLOS ONE

Dear Dr. Horiguchi,

I'm pleased to inform you that your manuscript has been deemed suitable for publication in PLOS ONE. Congratulations! Your manuscript is now being handed over to our production team.

Kind regards,

on behalf of

Dr Yalong Dang

Academic Editor

PLOS ONE